# Hepatitis B Virus preS/S Truncation Mutant rtM204I/sW196* Increases Carcinogenesis through Deregulated *HIF1A, MGST2,* and *TGFbi*

**DOI:** 10.3390/ijms21176366

**Published:** 2020-09-02

**Authors:** Ming-Wei Lai, Kung-Hao Liang, Chau-Ting Yeh

**Affiliations:** 1Liver Research Center, Chang Gung Memorial Hospital, Linkou Branch, Taoyuan 333, Taiwan; 2Division of Pediatric Gastroenterology, Department of Pediatrics, Chang Gung Memorial Hospital, Linkou Branch, Taoyuan 333, Taiwan; 3Molecular Medicine Research Center, College of Medicine, Chang Gung University, Taoyuan 333, Taiwan; 4Medical Research Department, Taipei Veterans General Hospital, Taipei 112, Taiwan; kunghao@gmail.com

**Keywords:** hepatitis B virus, reverse-transcriptase domain, drug-resistant mutant, preS/S truncation mutant, oncogenesis

## Abstract

Inevitable long-term therapy with nucleos(t)ide analogs in patients with chronic hepatitis B virus (HBV) infection has selected reverse-transcriptase (rt) mutants in a substantial proportion of patients. Some of these mutants introduce premature stop codons in the overlapping surface (s) gene, including rtA181T/sW172*, which has been shown to enhance oncogenicity. The oncogenicity of another drug-resistant mutant, rtM204I/sW196*, has not been studied. We constructed plasmids harboring rtM204I/sW196* and assessed the in vitro cell transformation, endoplasmic reticulum (ER) stress response, and xenograft tumorigenesis of the transformants. Cellular gene expression was analyzed by cDNA microarray and was validated. The rtM204I/sW196* transformants, compared with the control or wild type, showed enhanced transactivation activities for *c-fos*, increased cell proliferation, decreased apoptosis, more anchorage-independent growth, and enhanced tumor growth in mouse xenografts. X box-binding protein-1 (*XBP1*) splicing analysis showed no ER stress response. Altered gene expressions, including up-regulated *MGST2* and *HIF1A*, and downregulated transforming growth factor beta-induced (*TGFbi*), were unveiled by cDNA microarray and validated by RT-qPCR. The *TGFbi* alteration occurred in transformants with wild type or mutated HBV. The altered *MGST2* and *HIF1A* were found only with mutated HBV. The rtM204I/sW196* preS/S truncation may endorse the cell transformation and tumorigenesis ability via altered host gene expressions, including *MGST2, HIF1A,* and *TGFbi*. Downregulated *TGFbi* may be a common mechanism for oncogenicity in HBV surface truncation mutants.

## 1. Introduction

Hepatitis B virus (HBV) infection claims an enormous health burden on the globe, with over 250 million people living with chronic hepatitis B, and a significant proportion of them suffering from the consequences of cirrhosis and hepatocellular carcinoma. Since the approval of Lamivudine (3TC) as the first nucleoside analog (NA) in 1998 for the treatment of chronic hepatitis B virus (HBV) infection, this drug has been widely used around the world. One year of 3TC therapy in hepatitis e antigen (HBeAg)-positive patients may achieve alanine aminotransferases (ALT) normalization, HBeAg loss, and improved liver fibrosis in up to 70%, 30%, and 60% of patients, respectively [1]. The HBeAg seroconversion rate was enhanced from 17% to 50% after 2–5 years of extended use, but the treatment takes a toll, increasing drug-resistant mutants from 23 to 70% [2,3,4,5]. Subsequently, newer NAs with a higher potency to suppress HBV replication and higher genetic barriers to resistant mutations were approved. The benefit of NAs to reduce the risk of hepatocellular carcinoma (HCC) in chronic hepatitis B (CHB) patients is well documented [6,7,8,9]. However, the advantage may be offset if NA-resistant mutants emerge [6,10,11,12].

The resistant mutants mostly locate in the reverse-transcriptase (rt) B and C domains, which might result in simultaneous surface gene mutations because of overlapping reading frames, and some of them cause stop-codon truncation of envelope proteins [13]. In our previous study, we analyzed the pre-S/S sequences in HCC patients, in whom the liver cancer developed after lamivudine therapy. We discovered exclusive mutations resulting in premature stop codons in the S gene rather than variations in the pre-S region, compared to those HCC cases without lamivudine treatment. Nonsense surface gene mutations have also been identified in treatment-naïve HCC patients [14]. Truncated preS/S2 mutants were identified in the integrated HBV genome from hepatoma, and are capable of transactivating oncogenic proteins [15]. Transactivation activity occurred only when the premature stop codons were located within a region of the S gene, named the transactivity-on-region [16]. We investigated four such mutants (sL15*, sL21*, sW156*, and sW172*) in our 3TC-experienced HCC cases for their nearby locations to the transactivity-on-region, and proved sL15*, sL21*, and sW172* mutants to be oncogenic [14,17]. rtM204I is a common 3TC-resistant mutation emerging in antiviral therapy, and may result in truncation or missense mutations on the overlapping surface gene (sW196*/S/L). Notably, this truncation mutation was also found in Telbivudine- or Entecavir-experienced HCC patients [17,18]. Although NA with a high genetic barrier (Entecavir or Tenofovir) is recommended as first-line therapy, 3TC has been used for quite a long time, and is still in the market in some countries because of its lower cost. In this study, we proved the oncogenic potential of rtM204I/sW196* mutants via in vitro assays in order to assess the cellular transformation ability and mouse xenograft tumorigenesis. Furthermore, we explored the mechanism via cDNA microarray and identified altered host gene expressions, validated by RT-qPCR, in mutant-transformants.

## 2. Results

### 2.1. rtM204I/sW196* Resulted in an Enhanced Transactivation of Proto-Oncogenes, Increased Cell Proliferation, and Decreased Apoptosis, but Did Not Elicit an Endoplasmic Reticulum (ER) Stress Response

The rtM204I/sW196* (YIDDst) plasmid was constructed via a PCR-based site-directed mutagenesis method, and the sequence was confirmed by Sanger sequencing, as shown in Figure 1A. The expression of mutant HBsAg in the NIH3T3 cells was confirmed by immunofluorescence analysis, as shown in Figure 1B. To select a high expression stable transformant of the NIH3T3 cells, RT-qPCR was performed, and a high-expression transformant (clone 10) was used for the subsequent cell assays due to the loss of clone 12 (Figure 1C).

Simultaneous transfection of the pGL3-reporter plasmid of *c-fos*-promoter, *c-myc*-promoter, or SV40 with pIRES-YIDDst in NIH3T3 cells revealed enhanced transactivation of the *c-fos*-promoter and SV40 (*p* = 0.0077 and 0.0643, respectively, *t*-test), more than that of the pIRES-WT transformants, as shown in Figure 2A.

The pIRES-YIDDst-transfected cells showed increased cell proliferation by MTT assays on days 3, 5, and 7 compared with pIRESmock-transfected NIH3T3 cells (*p* = 0.0099, 0.0033, and 0.0250, respectively), or on days 3 and 5 compared with WT-transfected cells (*p* = 0.0009 and 0.0034, respectively). The WT transformants showed a lagging, but steadily increased proliferation from day 3 to day 7, despite not showing a significant difference with the YIDDst (*p* = 0.1795) or mock (*p* = 0.0621, *t*-test) transformants on day 7, as shown in Figure 2B.

The reverse transcription PCR method was applied to detect ER stress-specific X box-binding protein-1 (*XBP1)* mRNA splicing. The YIDDst Huh 7 transformants showed no spliced form, implying no evidence of ER stress response, as shown in Figure 2C.

Camptothecin-induced apoptosis of NIH3T3 transformants was assessed by fluorescein isothiocyanate (FITC)-labeled Annexin V via flow cytometry. Compared with the pIRESmock-transfected cells, the pIRES-YIDDst and WT transformants both showed significantly decreased apoptosis (29.6 ± 20.2%, 1.2 ± 0.8%, and 2.3 ± 1.1%, respectively), with the YIDDst transformants displaying the lowest apoptosis, as shown in Figure 3A,B.

### 2.2. rtM204I/sW196* Resulted in More Anchorage-Independent Growth and Enhanced Mouse Xenograft Tumorigenesis

Anchorage-independent growth was evaluated by colony formation in the soft agar cell culture. Compared with pIRESmock-transfected NIH3T3 cells or wild type transformants, the pIRES-YIDDst transformant had significantly increased colony counts in the soft agar (Table 1). Tumorigenesis in the xenograft mouse assay also showed an increased proportion and size of tumor growth in pIRES-YIDDst transfected NIH3T3 cells (Table 1 and Figure 4).

### 2.3. Genome-Wide Exploration of Expression Alterations in YIDDst-Transformed Cells

The cDNA microarray analysis for the NIH3T3 cells stably transfected with pIRES-YIDDst versus pIRES-mock revealed differentially expressed genes, as shown in the heatmap (Figure 5A). A total of 79 genes were significantly altered in expression, with 33 up-regulated and 46 downregulated in the YIDDst cells. Some of these genes could be ascribed to multiple aspects of the hallmarks of the cancer conceptual model [19], including sustaining proliferative signals (*HIF1A* (hypoxia-inducible factor 1 subunit alpha) and *HLF* (hepatic leukemia factor) up; *LEF* (lymphoid enhancer-binding factor), down), avoiding immune destruction (*Etv1* (ETS variant transcription factor 1), *PIP5K1A* (phosphatidylinositol-4-phosphate 5-kinase type 1 alpha) up), tumor-promoting inflammation (*PIP5K1A, SAR1b* (secretion associated Ras-related GTPase 1B), *MGST2* (microsomal glutathione S-transferase 2), *PGF2AR* (prostaglandin F 2 alpha receptor), up), activating invasion and metastasis (*HIF1A, Etv1,* up; *TGFbi* (transforming growth factor beta-induced), down), inducing angiogenesis (*HIF1A, HLF*, up), genome instability, and mutation (*MGST2*, up), and resisting cell death (*HIF1A*, up), as shown in Figure 5B. Based on the importance of multiple aspects of cancer, we then focused on three up-regulated genes, *MGST2, HIF1A*, and *Etv1*, and two downregulated genes, *TGFbi* and *LEF1*, for the subsequent validation.

### 2.4. Validation of the Differentially Expressed Transcripts in Transformants with WT, Mutant, or Mock-Control by RT-qPCR

RT-qPCR of the extracted-RNA from pIRESmock-, YIDDst-, and WT-transfected NIH3T3 cells was performed, and the expression was normalized to the actin levels. As shown in Figure 6A (after logarithm transformation and relative to the expression level of the mock-control cells), pIRES-YIDDst transformants had a significant up-regulated *MGST2* and downregulated *TGFbi* compared with the control cells (*p* = 0.0028 and 0.0001, respectively, multiple *t*-tests). YIDDst transformants also had significantly up-regulated expressions of *MGST2* and *HIF1A* compared with the WT transformants (*p* = 0.0020 and 0.0036, respectively). Up-regulated *HIF1A* and *Etv1* and downregulated *LEF1* expressions were observed in the YIDDst transformants, but not to a significant level (*p* = 0.0250, 0.3603, and 0.3100, respectively) compared with the control cells.

The expression of *TGFbi* may be induced in the presence of TGF-β. Therefore, we re-assessed the expression of *TGFbi* after adding TGF-β1. As shown in Figure 6B (after logarithmic transformation and relative to that without adding TGF-β1), YIDDst and WT transformants had no induced expression of *TGFbi* compared with the control ones (*p* = 0.0007 and 0.0013, *t*-test, respectively). The induction was lower in the YIDDst transformants than that of WT, yet only at a marginally significant level (*p* = 0.0557).

## 3. Discussion

Nucleos(t)ide analogs have been widely used in the treatment of chronic hepatitis B patients for decades, and have improved patients’ clinical outcomes. However, the infinite treatment course may provoke polymerase-resistant mutations with concomitant overlapping surface gene mutations, among which the C-terminal truncation mutants have an increased potential for hepatocarcinogenesis in clinical patients as well as in in vitro cell and mouse models [17,20,21,22]. Previously, we identified rtA181T/sW172* in HCC patients, either 3TC-treated or -naïve [14,17]. We also explored the oncogenesis mechanism in transgenic mice carrying the preS/S sW172* mutation [20]. Other researchers also identified sW182* in HCC tissues, and demonstrated its enhanced tumorigenesis and migration ability in vitro [21,22]. The downregulated *TGFbi* expression was revealed by cDNA microarray study of the sW182*-transfected cells due to the hypermethylation of its promoter [23]. As for rtM204I/sW196* (YIDDst in this study), a mutant less commonly encountered than rtM204I/sW196S/L (* 8.2%, L 76.4%, S 10.2%, and L/S 5.1%) in 3TC or Telbivudine-experienced patients [24], we proved that YIDDst-transfected cells presented a transformation and tumorigenesis potential. To unveil the oncogenic mechanisms, we employed the gene expression microarray for a genome-wide exploration. The YIDDst-cells showed differential gene regulation in multiple cancer-promoting sectors. We then focused on a few candidate genes using RT-qPCR, and confirmed a significant up-regulated *MGST2* (versus control or WT), *HIF1A* (versus WT), and downregulated *TGFbi* (versus control) in YIDDst-cells. Furthermore, the *TGFbi* expression showed no induction in the presence of TGF-β (versus control significantly, and versus WT with marginal significance). This finding coincided with that in the sW182* study by Jiang et al. [23].

The enhanced *MGST2* and *HIF1A* represent two effects specific to the mutant HBV in contrast to the WT HBV. *MGST2* encodes a protein that catalyzes the conjunction of reduced glutathione with leukotriene A4 to form leukotriene C4 (LTC4). It is highly expressed in the liver and overexpressed in HCC (medium in 27% and high in 55% HCC tissues) [25]. ER stress may provoke LTC4 biosynthesis by the transcriptional activation of *MGST2*. LTC4 may, in turn, cause the accumulation of reactive oxygen species and oxidative DNA damage, a culprit in cancer, including hepatocellular carcinoma [26,27]. Although we did not prove an ER stress response in terms of *XBP1* splicing in YIDDst-cells, up-regulated *MGST2* may directly enhance LTC4 bypassing ER stress. Apart from activated cyclooxygenase in many epithelia-derived malignancies, *MGST2* up-regulation may enhance the eicosanoids-mediated crosstalk between tumor cells and the surrounding stromal cells, facilitating a tumor microenvironment [28]. 

*HIF1A* encodes a vital regulator of the cellular and systemic response to hypoxia by triggering the transcription of many genes involving energy metabolism, angiogenesis, apoptosis, and so forth, to increase oxygen transport or enable metabolic adjustment to hypoxia. *HIF1A* thus plays an indispensable role in tumor angiogenesis, and is a well-recognized marker in many cancers, including HCC. Between 9~18% of HCC showed medium to high protein levels [29]. HBV X protein (HBx), a multifunctional protein and transcriptional activator, may stabilize *HIF1A* via enhancing von Hippel–Lindau protein ubiquitination [30]. Furthermore, several common HBx mutations may enhance *HIF1A* expression [31]. PreS/S truncation-related *HIF1A* up-regulation has never been reported, perhaps via its pleiotropic transactivating ability [32]. 

*TGFbi*, also known as *βig-H3*, encodes an RGD (Arg-Gly-Asp)-containing protein that binds to collagens in many extracellular matrix proteins and interacts with several integrins. The protein is induced by TGF-β, and acts to inhibit cell adhesion. Conflicting phenomena of either tumor suppressors or promoters have been linked to various cancers. The loss or downregulation of *TGFbi* (a tumor suppressor) is related to lung cancer, breast cancer, and ovarian cancers [33,34,35], whereas overexpression (a tumor promoter) is associated with pancreatic, colorectal, and other gastrointestinal malignancies [36,37,38]. The protein is lowly expressed in healthy liver tissue. About 70% of HCC showed low to undetectable levels of *TGFbi* [39]. C-terminal truncated surface gene mutations, including sW172*, sW182* [23,40], and sW196* in this study, coincidentally resulted in *TGFbi* downregulation in transfected-cells, which presented transforming characteristics and mouse xenograft tumorigenesis.

## 4. Materials and Methods

### 4.1. Plasmid Construction and Site-Directed Mutagenesis

The wild type preS/S region of HBV was amplified from pCMV-HBV, which contained one copy of greater-than-unit-length HBV genome (3.37 kb; nt. 1820 to 2990; adw subtype; GenBank accession number, X02763), using PS1: 5′-ATATTCTTGG GAACAAGAGC-3′ (nt. 28282847–, sense) and PS2: 5′-GGAATAACCCCATCT TTTTG-3′ (nt. 867848–, anti-sense) as primers. The PCR product was blunt-ended and inserted into pRC/CMV (Invitrogen, San Diego, CA, USA) to generate pCMV-preS/S-WT (wild type). Site-directed mutagenesis was performed to introduce the sW196* (YIDDst, YIDD/S truncation) mutation via a PCR-based method described previously [12]. A set of primers, complementary to each other and spanning the mutation sites, YIDDs (5′-GGCTTTCAGCTATATAGATGATGTGG-3′) and YIDDas (5′-CCACATCATCTATATAGCTG AAAGCC-3′), were synthesized. Two sets of PCRs (30 cycles) were performed using PS1/YIDDas and PS2/YIDDs as PCR primers, respectively. The resulting DNA fragments were gel purified. One-tenth of each purified DNA fragment was mixed, and another PCR reaction (10 cycles) was performed in the absence of primers. Finally, PS1 and PS2 were added into the reaction, and 20 more cycles of PCR were performed. The resulting DNA fragment, containing the desired mutation, was blunt-ended and inserted into pRC/CMV to generate pCMV-preS/S-196* (or pCMV-YIDDst). PCR products were inserted into pIRESbleo (BD Biosciences Clontech, Palo Alto, CA) instead of pRC/CMV. The resulting plasmid was pIRES-preS/S-WT (WT) and pIRES-preS/S-196* (YIDDst), and these plasmids were used to establish stable transformants in the cells. All of the plasmids were sequence-verified using the automatic DNA sequencer. 

### 4.2. Cell Culture, Plasmid Transfection, and Immunofluorescence Analysis 

Huh7 cells and NIH3T3 cells were maintained in Dulbecco’s modified Eagle’s medium containing 10% fetal calf serum. Transfection was performed using Lipofectamine 2000 reagent (Invitrogen, Carlsbad, CA). Stable clones were selected by G418, and the expression levels of eleven clones were assayed by reverse transcription qPCR.

For immunofluorescence staining, NIH3T3 cells were grown on cover slides and transfected with pIRES-YIDDst. Forty-eight hours after the transfection, the cells were fixed in acetone at −20 °C for 2 min. A rabbit polyclonal anti-HBs antibody (ViroStat; 1:100 dilution) and a fluorescein isothiocyanate-conjugated goat anti-rabbit antibody (Leinco Technologies, Inc., Saint Louis, MO, USA; 1:150 dilution) were used as the primary and secondary antibodies, respectively. The cells were stained with 4′,6-diamidino-2-phenylindole (DAPI; 200 ng/mL) to visualize the nuclei.

### 4.3. Transactivation Analysis with Dual-Luciferase Assay 

To construct pGL3-cFos-P and pGL3-cMyc-P, the promoter areas of *c-Fos* and *c-Myc* were amplified from chromosomal DNA. The primers were PF1 (5′-GCTCGAGCCCGAGGGCTGGAG-3′; nt. 21–41, sense) and PF2 (5′-TGCGGTTGG AGTACGAGGCG-3′; nt. 740–721, anti-sense) for the *c-Fos* promoter and PM1 (5′-GCGCTCTCCAAGTATACGTGGC-3′; nt. 29–50, sense) and PM2 (5′-CAGCGAGTTA GATAAAGCCC-3′; nt. 818–799, anti-sense) for the *c-Myc* promoter. The resulting DNA fragments were blunt-ended and inserted into pGL3-Basic (Promega, Madison, WI) upstream of the luciferase gene. The plasmid, pGL3-SV40-P, was the same as the pGL3-Promoter (Promega), which contained the SV40 promoter sequence upstream of the luciferase gene. The plasmid, pGL4.73 (hRluc/SV40; Promega) was used as an internal control. Cotransfections of pCMV-YIDDst, -WT, and the mock control with pGL3-cFos-p, pGL-cMyc-p, and pGL3-SV40-p, respectively, and pGL4.73 (hRluc/SV40) in a 10:1:0.2 ratio were performed with Lipofectamine 2000. After cotransfection for 30 h, the cells were lysed and analyzed with the Dual-Luciferase Reporter assay system (Promega). Fold increase of activation was calculated by comparison of the luciferase activity between pCMV-YIDDst- or pCMV-WT- and mock-transfected cells.

### 4.4. Cell Proliferation Assay

Cell proliferation was assessed by using the 3-(4, 5-dimethylthiazol-2-yl)-2,5-diphenyltetra- zolium bromide (MTT) assay. In summary, NIH3T3 cells at a cell density of 2 × 10^3^ per well were cultured in 96-well plates for 1 to 7 days. On the day of measurement, the cells were incubated with the MTT solution (sigma, 0.5mg/mL) for 4 h at 37 °C. The cells were lysed with Dimethyl Sulfoxide (DMSO), and the formazan salt was solubilized under gentle agitation for 30 min. The absorbance was measured at 570 nm with a microplate reader. 

### 4.5. FITC Annexin V Apoptosis Assay by Flow Cytometry

NIH3T3 cells (2 × 10^5^ cells) were plated in six well culture dishes and treated with five µM camptothecin (Sigma-Aldrich, Inc. Saint Louis, MO, USA) for 6 h to induce apoptosis. The cells were harvested by 0.5% trypsin (Gibco) and collected in a 3-mL tube and centrifuged at 1000 rpm for 5 min. After washing the cells twice with cold phosphate-buffered saline, the cells were resuspended in a 200 µL 1× binding buffer. Then, 100 µL of the solution was transferred to a 3-mL culture tube, and 5 µL fluorescein isothiocyanate (FITC)-Annexin V and Propidium Iodide (BD Pharmingen, San Jose, CA, USA) was added and incubated for 15 min at room temperature in the dark. Finally, 400 µL of 1× binding buffer was added to each tube and was analyzed by flow cytometry (Beckman Coulter, Fullerton CA, USA).

### 4.6. Assessment of ER Stress Response

To assess endoplasmic reticulum (ER) stress response in the Huh 7 stable transformants, spliced and unspliced forms of human X box-binding protein-1 (*XBP1*) mRNA were detected by RT-PCR. The primers used were 5′-GAACCAAAAACTTTTGCTAG-3′ (sense, nt. 332-351, GenBank accession number, AB076383) and 5′-AGGATATCAGACTCTGAATCT-3′ (antisense, nt. 618-598). The cells treated by brefeldin A (1 µg/mL) for 24 h before assay were used as a positive control. 

### 4.7. Anchorage-Independent Growth by Soft Agar Assay

Two thousand transfected NIH3T3 cells in 2 mL of cell culture medium containing 10% FCS with 0.3% agar were laid on a 1-ml basal layer of 0.8% wt/v agar in 35 mm plates. Fresh medium was added every week. Colony formation was determined on day 21. Three repeats were done for each transformant.

### 4.8. Mouse Xenograft Tumorigenicity Assay

The protocol and procedures employed for the mouse experiments were ethically reviewed and approved by the Laboratory Animal Center, Chang Gung Memorial Hospital, Linkou on 30 December 2008 (no. 2008121509). Transfected NIH3T3 cells were harvested by trypsinization, washed twice with sterile phosphate-buffered saline, and resuspended at 1 × 10^7^ cells per ml. Aliquots (0.1 mL) were injected subcutaneously into 6-to 8-week-old Balb/c nude mice. Mice were observed periodically for tumor formation for 15 weeks.

### 4.9. cDNA Microarray

Three pairs of pIRES-YIDDst and pIRESmock-transfected NIH3T3 cells were collected for complementary DNA (cDNA) microarray analysis. The total RNA was isolated from the cells using Trizol reagent (Invitrogen Corporation, Carlsbad, CA, USA). The cDNA microarray system used in this study was GeneChip Gene 1.0 ST Array System for Mouse by Affymetrix, which interrogated 28,853 well-annotated genes with 770,317 distinct probes. 

### 4.10. Real-Time RT-PCR Validation of Up- and Down-Regulated Genes

The highest or critical up- and down-regulated genes from cDNA microarray analysis were validated by quantitative reverse transcription PCR (RT-qPCR). Stable transformants of NIH3T3 cells were collected for RNA extraction. The first-strand cDNA was synthesized from 0.5 μg of total RNA using SuperScript III First-Strand Synthesis System (Invitrogen, Carlsbad, CA, USA), followed by PCR using the SYBR Green PCR Master Mix (Life Technologies Co., Carlsbad, CA, USA). The primers used for RT-qPCR are listed in Table 2. For validation of a TGF-β induced expression of *TGFbi*, transfected cells were cultured with or without TGF-β1 (ProSpec-Tany TechnoGene, Israel) 0.5 nM treatment for 7 h after serum starvation for 16 h and followed by RNA extraction and RT-qPCR assay.

### 4.11. Statistical Analysis

Except for cDNA microarray data analysis, all of the other statistical comparisons were conducted by Student’s *t*-test for continuous data and by Fisher’s exact test for categorical data. Logarithm transformation was applied in the assays with skewed data. A *p*-value of <0.05 was considered significant when a single comparison was made, and of <0.01 when multiple comparisons were made. The methodology of the cDNA microarray data analysis has been described previously [20].

## 5. Conclusions

The HBV preS/S C-terminal truncation mutant sW196*, which may be selected in NA (3TC, Telbivudine, or Entecavir)-resistant rtM204I strains, potentially confers cell transformation and tumorigenesis ability through altered (up- or down-regulated) cellular transcriptions that orchestrate in different cancer-promoting sectors. Downregulated *TGFbi* not only contributes to oncogenicity in sW196*-transfected cells, but also in other preS/S* (sW172* and sW182*) cells, suggesting a common oncogenic mechanism shared by the surface truncation mutants; some may emerge during NA antiviral therapy.

## Figures and Tables

**Figure 1 ijms-21-06366-f001:**
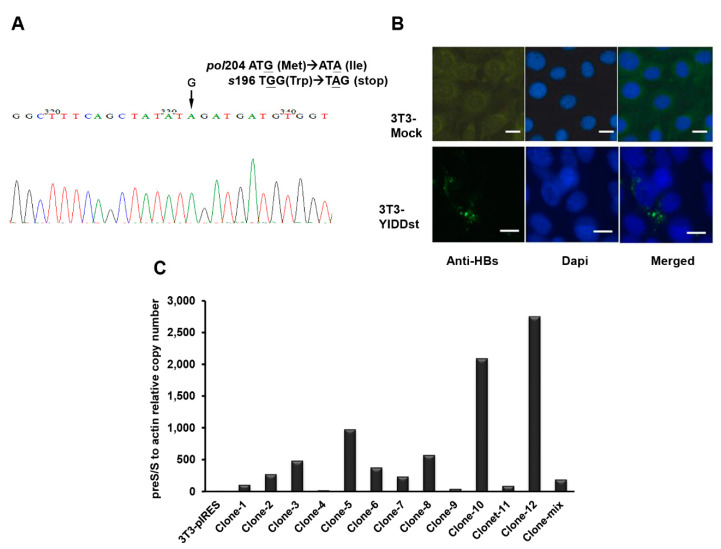
Plasmid construction and cellular expression of the rtM204I/sW196* mutant. (**A**) Sanger’s sequencing confirms the introduction of the stop codon (TGG > TAG) by site-directed mutagenesis at the surface protein 196th amino acid (W > *) and concomitant rt 204 M > I in the plasmids carrying rtM204I/sW196* (YIDDst). (**B**) Immunofluorescent staining of YIDDst-transfected NIH3T3 cells with fluorescein isothiocyanate (FITC)-conjugated antibody to HBsAg (lower panel) versus mock control cells (upper panel). DAPI: 4′,6-diamidino-2-phenylindole for nuclear counterstain. Scale bar: 20 μm. (**C**) Real-time RT-PCR of multiple stable clones of pIRES-YIDDst transfected NIH3T3 cells. A higher expressive clone (#10) was used for the subsequent assays.

**Figure 2 ijms-21-06366-f002:**
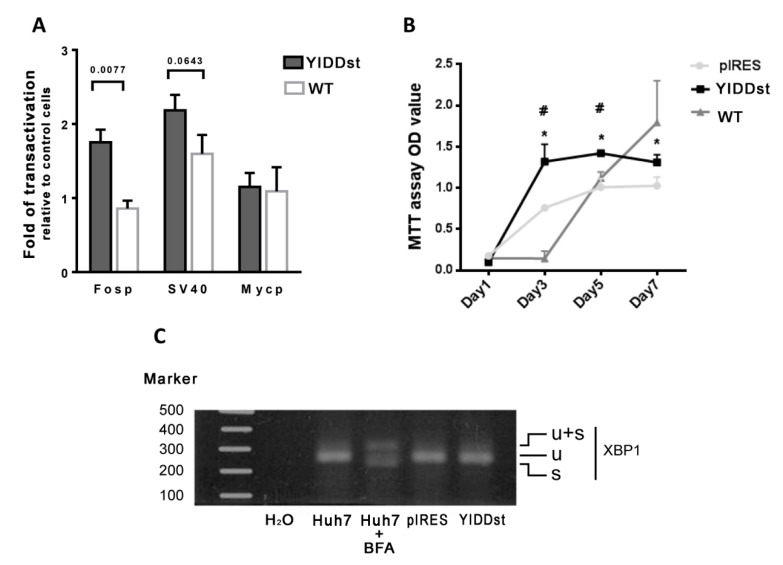
Assays of the transactivation, cell proliferation, and endoplasmic reticulum (ER) stress response. (**A**) Fold change of the dual-luciferase reporter assay relative to the mock control shows a higher transactivation of *c-fos* and SV40 in YIDDst than wild type (WT) cotransfection. (**B**) Cell proliferation with 3-(4, 5-dimethylthiazol-2-yl)-2,5-diphenyltetrazolium bromide (MTT) assay shows a significantly increased proliferation of YIDDst-cells compared with that of the mock control cells (* *p* = 0.0099, 0.0033, and 0.0250 on days 3, 5, and 7, respectively, *t*-test) and with that of the WT cells (# *p* = 0.0009 and 0.0034 on days 3 and 5, respectively, *t*-test). (**C**) X box-binding protein-1 (*XBP1*) mRNA splicing assay shows a lack of ER stress response in YIDD-cells. BFA—brefeldin A, added as an ER stress inducer; u—unspliced; s—spliced; u + s—unspliced and spliced hybrid.

**Figure 3 ijms-21-06366-f003:**
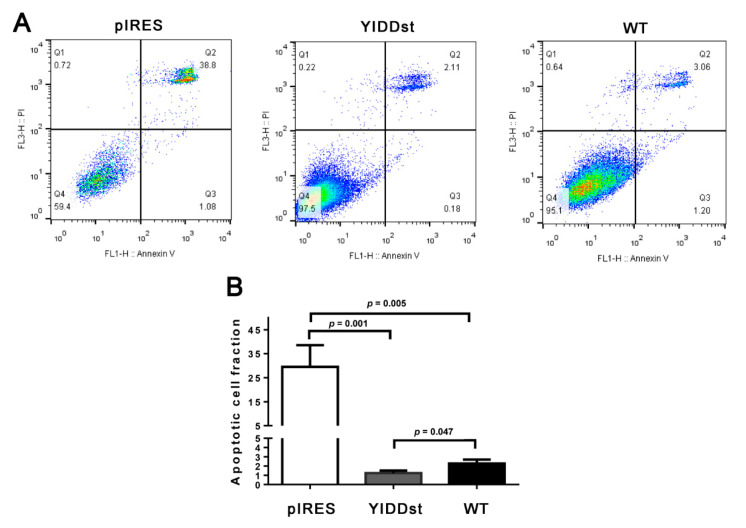
Apoptosis assay. (**A**) The rrepresentative Annexin V and Propidium Iodide (PI) flow cytometry shows significantly less camptothecin-induced apoptosis (Q2 + Q3) in the YIDDst and WT cells than the mock control-cells. (**B**) The bar chart shows the quantitative data of five individual assays; statistical analysis done using *t*-test.

**Figure 4 ijms-21-06366-f004:**
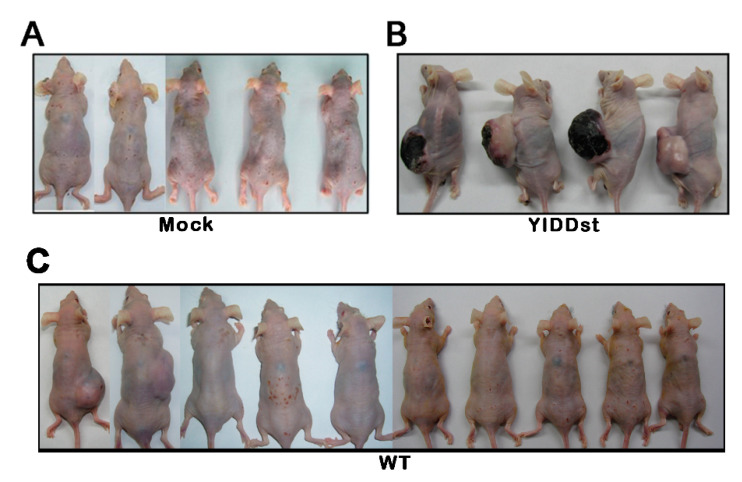
Mouse xenograft tumorigenesis. 1 × 10^6^ cells were injected subcutaneously and observed for 15 weeks. (**A**) pIRESmock-transfected cells, (**B**) pIRES-YIDDst cells, and (**C**) pIRES-WT cells.

**Figure 5 ijms-21-06366-f005:**
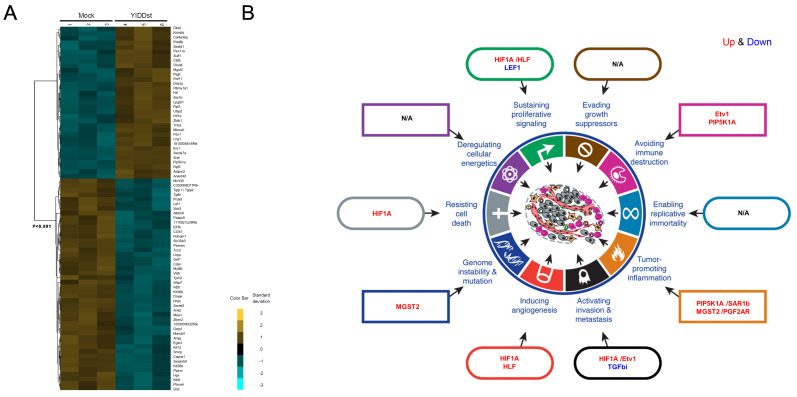
Differentially expressed genes in YIDDst-cells. (**A**) The heatmap after analysis by Cluster 3.0 and TreeView shows the panels of up- and down-regulated genes in three pairs of YIDDst- and mock control-transfected NIH3T3 cells. (**B**) Linking the critically differentially expressed genes to the hallmarks of cancer; up- and down-regulated (in red and blue, respectively) genes are annotated into different sectors of the conceptual model. Figure B was adapted from the publisher with permission.

**Figure 6 ijms-21-06366-f006:**
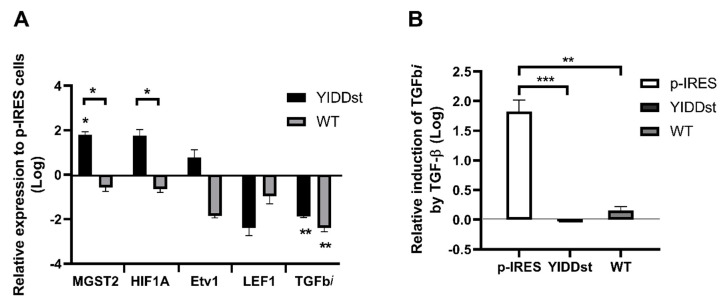
Validation of differentially expressed genes by RT-qPCR. (**A**) The relative expressions of five genes in the YIDDst- or WT-transfected cells (black and grey bar, rspectively) to the mock-control cells are shown. * *p*< 0.01; ** *p* < 0.001, multiple *t*-test; the n-zig-zag line shows significant differences between the expressions in the YIDDst- and WT-cells. (**B**) The relative induction of transforming growth factor beta-induced (*TGFbi*) after the addition of TGF-β in the control-, YIDDst-, and WT-cells. ** *p* < 0.01, *** *p* < 0.001, *t*-test.

**Table 1 ijms-21-06366-t001:** Anchorage-independent growth and xenograft tumorigenesis in NIH3T3 transformants.

Transformants	Colony Counts in Soft Agar	Tumor Growth Proportion	Tumor Size (cm^3^)
**pIRES**	0.33 ± 0.58 *******^,^*****	0/5 **^‡^**^,^**^§^**	-
**pIRES-YIDDst**	101.07 ± 28.87 *******^,^**^†^**	4/4 **^††^**^,^**^‡^**^,^**^¶^**	5.95 ± 1.63 ^#^
**pIRES-WT**	49.67 ± 5.51 *****^,^**^†^**	2/10 **^§^**^,^**^¶^**	2.37 ± 0.03 ^#^

*******^,^***********^,^****^†^** statistics by *t*-test, *****
*p* = 0.038, *******
*p* = 0.0001, **^†^**
*p* = 0.039. **^‡^****^,^****^§^****^,^****^¶^** statistics by Fisher’s exact test, **^‡^**
*p* = 0.079, **^§^**
*p* = 0.523, **^¶^**
*p* = 0.015. **^#^** statistics by *t*-test, *p* = 0.043. **^††^** a mouse died early because of hydronephrosis.

**Table 2 ijms-21-06366-t002:** Primer sets designed for reverse-transcription qPCR in the validation study.

Symbols	Gene Title	Forward Primer	Reverse Primer
*MGST2*	Microsomal glutathione S-transferase 2	GCCATCCACAGCATTACGAT	AAAACACAAGATTGCACCCC
*HIF1A*	Hypoxia-inducible factor 1-alpha	AAGTGGCAACTGATGAGCAA	GGCGAGAACGAGAAGAAAAA
*Etv1* *(ER81)*	Ets Variant Gene 1 (Ets-Related Protein 81)	CTTGATTTTCAGTGGGAGGC	AACATGGCTTGCTGAAGCTC
*LEF1*	Lymphoid Enhancer Binding Factor 1	AGAAAAGTGCTCGTCGCTGT	AAATGGGTCCCTTTCTCCAC
*TGFbi*	Transforming growth factor beta-induced	CCTTTCTCTCCTGGGACCTT	TCATTGGCACCAACAAGAAA

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
