# Peer review of "Hepatitis B Virus preS/S Truncation Mutant rtM204I/sW196* Increases Carcinogenesis through Deregulated HIF1A, MGST2, and TGFbi"

_ijms, 2020, doi:10.3390/ijms21176366_

Round 1

Reviewer 1 Report

The aim of the presented paper is to prove the oncogenic potential of drug-resistant mutants as a result of long-term therapy with nucleos(t)ide analogs in patients with chronic hepatitis B virus. The authors have proved the oncogenic 61 potential of rtM204I/sW196* mutants via in vitro assays to assess the cellular transformation ability 62 and mouse xenograft tumorigenesis. Some of these mutants introduce premature stop codons in the overlapping surface (s) 15 gene, including rtA181T/sW172*, which has been shown to enhance oncogenicity. Material and Methods are clearly described and sufficiently explained. The material is very interesting, original and significant. There will be an overall benefit to publishing this work. The work provides an up-to-date knowledge of the long-term therapy with nucleos(t)ide analogs in patients with chronic hepatitis B virus. The English language is appropriate and understandable.

Author Response

The aim of the presented paper is to prove the oncogenic potential of drug-resistant mutants as a result of long-term therapy with nucleos(t)ide analogs in patients with chronic hepatitis B virus. The authors have proved the oncogenic potential of rtM204I/sW196* mutants via in vitro assays to assess the cellular transformation ability 62 and mouse xenograft tumorigenesis. Some of these mutants introduce premature stop codons in the overlapping surface (s) gene, including rtA181T/sW172*, which has been shown to enhance oncogenicity. Material and Methods are clearly described and sufficiently explained. The material is very interesting, original and significant. There will be an overall benefit to publishing this work. The work provides an up-to-date knowledge of the long-term therapy with nucleos(t)ide analogs in patients with chronic hepatitis B virus. The English language is appropriate and understandable.

Response: The authors thank the reviewer for appreciating our work and supporting it for publication in the International Journal of Molecular Sciences.

Reviewer 2 Report

The paper by Lai, Liang and Yeh presents a study in which they constructed plasmids to carry a rtM2041/sW196* mutant in the hepatitis preS/S. This mutation is believed to be generated as a result of anti-viral therapy against chronic hepatitis B virus, such as lamivudine, telbivudine and entecavir. In a previous report, they evaluated the oncogenic potential of a rtA181T/sW172* mutant. In the present paper, they investigated the cell transformation effect of this mutant in NIH3T3 cells. The authors report the mutant can transform NIH3T3 cells to increase cell proliferation and resistance to apoptosis. Further, using a cDNA microarray approach, they identified that MSTG2 and HIF1A were some of the genes whose expression was altered and may contribute towards the carcinogenesis mechanism. Hepatitis B virus infection is an established public health problem and chronic therapy has been shown to induce resistance by mutational changes in the viral genome. This problem is acknowledged but this reviewer has major concerns with the study. NIH3T3 cells are an immortalized fibroblast cell line and this reviewer finds it problematic that the authors used these cells to investigate transformation ability of the rtM2041/sW196* mutant in this cell line. This cell line has been used for similar transformation studies, but it is not a relevant model for drawing conclusions that are applicable to human hepatocellular carcinomas. HuH7 cells (a hepatic carcinoma cell line) was mentioned only once in the methods section, but nowhere else in the text. It is unclear what these cells were used for in this study. The authors must include relevant primary cells to demonstrate importance of the cell-transformation capacity of the mutant under study. The cDNA microarray was descriptive and additional studies are required to unravel the mechanism by which the rtM2041/sW196* mutant may be mediating cell transformation-this may be useful for mitigating carcinogenesis as these mutants are discovered.

Specific comments

The introduction should start by introducing hepatitis B virus and its impact on public health. This is important for the reader to appreciate the extent to which HBV is a public health problem.

The results are presented in brief subtitles, Sections 2.1-2.5 should be consolidated into 1 section.

Line 103: what do the authors mean by “stronger”?

Line 305: Is there a study approval number or ID after the approval?

Line 307: 107?

Figure 1B

There is background staining in the control cells and similar background staining is not seen in the transformed cells. Please explain. The figure does not have scale bars and the nuclei in the 3T3-YIDDst appear larger than in the controls. Were these images captured at the same time.

Figure 1C

The authors selected clone 10, clone 12 was higher than clone 10. Why was this not chosen? This clone should have been selected and compared to clone 10 to show that they behave similarly.

Figure 2B

The WT lags behind, but it causes increased cell proliferation from day 3 towards day 5. The subtitle 2.3 in the text maybe misleading and a clearer interpretation of the result is required.

Figure 2C

The authors indicate 2 band sizes depicting whether splicing was present or not. An apparent 3rd band size in lane

Figure 3

The authors use annexin 5 to show that the YIDDst ad the WT construct reduce sensitivity of NIH3T3 cells to apoptosis. How is the resistance to apoptosis mediated? What is the mechanism of resistance? Are the caspases cleaved, but the end result blocked or there is no cleavage at all? The authors need to exogenously induce apoptosis with a different compound known to provide additional support for the apoptosis resistance of transformed cells.

Figure 5A

The text is not readable, please enlarge.

Section 2.8: the title should be validation of array data with qPCR rather than “investigation”

Author Response

General comments

The paper by Lai, Liang and Yeh presents a study in which they constructed plasmids to carry a rtM2041/sW196* mutant in the hepatitis preS/S. This mutation is believed to be generated as a result of anti-viral therapy against chronic hepatitis B virus, such as lamivudine, telbivudine and entecavir. In a previous report, they evaluated the oncogenic potential of a rtA181T/sW172* mutant. In the present paper, they investigated the cell transformation effect of this mutant in NIH3T3 cells. The authors report the mutant can transform NIH3T3 cells to increase cell proliferation and resistance to apoptosis. Further, using a cDNA microarray approach, they identified that MSTG2 and HIF1A were some of the genes whose expression was altered and may contribute towards the carcinogenesis mechanism. Hepatitis B virus infection is an established public health problem and chronic therapy has been shown to induce resistance by mutational changes in the viral genome. This problem is acknowledged but this reviewer has major concerns with the study. NIH3T3 cells are an immortalized fibroblast cell line and this reviewer finds it problematic that the authors used these cells to investigate transformation ability of the rtM2041/sW196* mutant in this cell line. This cell line has been used for similar transformation studies, but it is not a relevant model for drawing conclusions that are applicable to human hepatocellular carcinomas. HuH7 cells (a hepatic carcinoma cell line) was mentioned only once in the methods section, but nowhere else in the text. It is unclear what these cells were used for in this study. The authors must include relevant primary cells to demonstrate importance of the cell-transformation capacity of the mutant under study. The cDNA microarray was descriptive and additional studies are required to unravel the mechanism by which the rtM2041/sW196* mutant may be mediating cell transformation-this may be useful for mitigating carcinogenesis as these mutants are discovered.

Response: The authors thank for the reviewer’s in-depth review and valuable suggestions. Most of the experiments for detecting transforming ability of overexpressed and/or mutated genes in tumors (oncogenes) are performed using mouse embryonic fibroblasts, NIH3T3 mouse fibroblast cell line, or human embryonic kidney 293 cell line (HEK293). These immortalized cells do not carry normal diploid karyotype. However, they cannot grow over one another in cell culture (contact inhibition), do not form colonies in soft agar (anchorage-dependent growth), and do not form tumors when injected into immunodeficient mice. In our experiments, we did transfect control and mutant plasmids into Huh 7 cells and perform soft agar assay and mouse tumorigenesis. The Huh7-mock grew colonies in soft agar and developed xenograft tumors in nude mice with an efficiency not significantly different from that of Huh7-YIDDst cells. Therefore, the sW196* mutant did not seem to perturb tumor growth once the cells had been transformed (by other mechanisms). Here, the authors showed the results using NIH3T3 cells (wherein no colonies and no xenograft tumor can be formed in mock-transfected cells) to assess the transformation ability of the YIDDst mutant. Huh7 cells presented the same finding in the XBP-1 splicing experiment as NIH3T3 cells (as shown in the revised Figure 2 and Materials and methods 4.6). As for your suggestion to use primary cells (like primary hepatocytes) in transformation experiments, the authors agree that it is an ideal cell model. Still, the primary cells need some genetic transduction to immortalize them (yet non-transformed) before the transformation assay can be done (e.g., Human NeHepLxHT immortalized hepatocytes (IMH1), a diploid human cell line that was developed by transduction with a retroviral vector containing the hTERT gene (ATCC cat no. CRL-4020)). As for cDNA microarray results, we aimed to search for genes with altered expressions in YIDDst transformant and to validate the findings. Just as your suggestion that additional studies are required to unravel the mechanism by which the rtM2041/sW196* mutant may be mediating cell transformation, we’ll design the next project for this purpose.

Specific comments

  1. The introduction should start by introducing hepatitis B virus and its impact on public health. This is important for the reader to appreciate the extent to which HBV is a public health problem.

Response: The authors have added a sentence to highlight the importance of hepatitis B virus infection on global health. Line 36-38, “Hepatitis B virus (HBV) infection claims an enormous health burden on the globe, with over 250 million people living with chronic hepatitis B and a significant proportion of them suffering from consequences of cirrhosis and hepatocellular carcinoma.” was added.

  1. The results are presented in brief subtitles, Sections 2.1-2.5 should be consolidated into 1 section.

Response: Sections 2.1-2.5 have been consolidated into one section “2.1. PreS/S truncation mutant rtM204I/sW196* resulted in enhanced transactivation of proto-oncogenes, increased cell proliferation, decreased apoptosis, but did not elicit an ER stress response”.

  1. Line 103: what do the authors mean by “stronger”?

Response: The authors intended to express both the higher proportion of tumor growth and larger tumor size in YIDDst-transfected cells compared to mock-control and WT-cells. Here (revised version, Line 115, subtitle 2.2.), we replaced “stronger” with “enhanced” mouse xenograft tumorigenesis.

  1. Line 305: Is there a study approval number or ID after the approval?

Response: The Animal Experiment Approval number was 2008121509, which has been added to the content (Line 320~321)

  1. Line 307: 107?

Response: Thank you for your correction. The error has been corrected to 107 (Line 322).

Figure 1B

There is background staining in the control cells and similar background staining is not seen in the transformed cells. Please explain. The figure does not have scale bars and the nuclei in the 3T3-YIDDst appear larger than in the controls. Were these images captured at the same time.

Response: Yes, the images were not captured on the same day. The background staining of the control cells was due to the Imager software setting to show the cytoplasm. In contrast, the anti-HBs immunofluorescence in the cytoplasm of YIDD cells indirectly outlined the cytoplasm without background staining. The scale bars were added in the images to show different conditions of magnification.

Figure 1C

The authors selected clone 10, clone 12 was higher than clone 10. Why was this not chosen? This clone should have been selected and compared to clone 10 to show that they behave similarly.

Response: We initially selected clone 12 for subsequent experiments, but the clone 12 cells were lost due to storage (frozen) failure, as explained on Line 80.

Figure 2B

The WT lags behind, but it causes increased cell proliferation from day 3 towards day 5. The subtitle 2.3 in the text maybe misleading and a clearer interpretation of the result is required.

Response: Yes, as the reviewer’s observation, the WT had a lagging but steadily increased cell proliferation from day 3 to 7. However, the difference was not significant between WT and YIDD (p= 0.1796, t-test) or between WT and mock (p= 0.0621, t-test) on day 7. The text has been rewritten to reflect the proliferative ability of WT transformants truthfully, as “The WT transformants showed a lagging but steadily increased proliferation from day 3 to day 7 despite not to a significant difference than YIDD (p= 0.1795, t-test) or Mock (p= 0.0621) transformants on day 7” on Line 92 to 94.

Figure 2C

The authors indicate 2 band sizes depicting whether splicing was present or not. An apparent 3rd band size in lane.

Response: The 3rd band is the hybrid of unspliced and spliced mRNA of XBP1. Thus, the lane labels were updated to describe the three lanes on the right of Figure 2 C, and the abbreviations (u, s, and u+s) were depicted in the caption. Besides, the authors were sorry about mislabeling the cell type used in this assay; Huh 7 cells were shown instead of NIH3T3 cells. The cell type label was corrected in Figure 2C, and the Materials and Methods section 4.6. (Line 307-311) was amended as well.

Figure 3

The authors use annexin 5 to show that the YIDDst and the WT construct reduce sensitivity of NIH3T3 cells to apoptosis. How is the resistance to apoptosis mediated? What is the mechanism of resistance? Are the caspases cleaved, but the end result blocked or there is no cleavage at all? The authors need to exogenously induce apoptosis with a different compound known to provide additional support for the apoptosis resistance of transformed cells.

Response: Loss of cellular membrane asymmetry is considered an early sign of apoptosis and persistent to cell death, commonly detected by Annexin V binding to phosphatidylserine by flow cytometry, as in our experiment. Early apoptotic cells are annexin V-positive and PI-negative (annexin V-FITC+/PI−, Q3), whereas late (end-stage) apoptotic cells are annexin V/PI-double-positive (Annexin V-FITC+/PI+, Q2). Caspases activation, including caspase 8 in the extrinsic pathway, and caspase 9 in the intrinsic path, followed by caspases 3/7 activation, the committed road leading to cell death, is another apoptotic marker (the authors did not choose this marker in this study). From the significantly less Q2 (late apoptotic) proportion in the YIDD and WT cells, resistant to caspase activation is speculated. The authors did not directly explore the mediators behind this phenomenon. However, in the cDNA microarray exploration of up- and down-regulated genes in the YIDD cells, HIF1A upregulation may contribute to resistance to cell death. The authors used camptothecin as an apoptosis-inducer with adequate apoptosis in the control cells and quite a significant difference in the YIDD or WT cells. The author had used Actinomycin D and TNF-α as an apoptosis inducer in cells harboring other preS/S truncation mutants before (Ref 17). Still, the authors applied a different marker and inducer for apoptosis in this study.

Figure 5A

The text is not readable, please enlarge.

Response: The resolution of Figure 5A was improved.

Section 2.8: the title should be validation of array data with qPCR rather than “investigation”

Response: Section 2.8 (2.4. in the revised version), the title was changed to “validation” of array data.

Reviewer 3 Report

Ming-Wei Lai and colleagues reported a study to investigate an HBV preS/S truncation mutant, i.e the rtM204I/sW196* that introduces a premature stop codon within the overlapping surface (s) HBV gene in parallel with the well-studied rtM204I mutant to assess the impact as a carcinogenesis factor and its potential contribution to progression to HCC for CHB carriers. In brief, the authors constructed plasmids harboring rtM204I/sW196* and assessed in vitro cell transformation, endoplasmic reticulum (ER) stress response, and xenograft tumorigenesis. Cellular gene expression was additionally assessed by cDNA microarray and validated by RT-qPCR for specific genes to validate their up and/or down regulation. The authors report enhanced transactivation activities for c-fos, increased cell proliferation, decreased apoptosis, more anchorage-independent growth, and enhanced tumor growth in mouse xenograft for the rtM204I/sW196* transformants, compared to the control or wild-type. They also confirmed up-regulated expressions for MGST2 and HIF1A genes and downregulation of the TGFbi, that may reveal a putative oncogenic mechanism in HBV surface truncation mutants. Overall, the ms reports a nicely designed and well-presented study that presents novelty and interesting data on a newly presented oncogenic mechanism that may contribute to the development of HCC for the rtM204I/sW196* mutant with a premature stop codon in the overlapping surface (s) gene.

Minor comment and suggestion

  • Figure’s 5 quality and size are too low, consider to replace with a bigger one that also has a better quality/resolution to reveal figure’s detailed infos.

Author Response

Ming-Wei Lai and colleagues reported a study to investigate an HBV preS/S truncation mutant, i.e the rtM204I/sW196* that introduces a premature stop codon within the overlapping surface (s) HBV gene in parallel with the well-studied rtM204I mutant to assess the impact as a carcinogenesis factor and its potential contribution to progression to HCC for CHB carriers. In brief, the authors constructed plasmids harboring rtM204I/sW196* and assessed in vitro cell transformation, endoplasmic reticulum (ER) stress response, and xenograft tumorigenesis. Cellular gene expression was additionally assessed by cDNA microarray and validated by RT-qPCR for specific genes to validate their up and/or down regulation. The authors report enhanced transactivation activities for c-fos, increased cell proliferation, decreased apoptosis, more anchorage-independent growth, and enhanced tumor growth in mouse xenograft for the rtM204I/sW196* transformants, compared to the control or wild-type. They also confirmed up-regulated expressions for MGST2 and HIF1A genes and downregulation of the TGFbi, that may reveal a putative oncogenic mechanism in HBV surface truncation mutants. Overall, the ms reports a nicely designed and well-presented study that presents novelty and interesting data on a newly presented oncogenic mechanism that may contribute to the development of HCC for the rtM204I/sW196* mutant with a premature stop codon in the overlapping surface (s) gene.

Response: The authors thank the reviewer for agreeing with our results.

Minor comment and suggestion

  • Figure’s 5 quality and size are too low, consider to replace with a bigger one that also has a better quality/resolution to reveal figure’s detailed infos.

Response: Thank you for your valuable suggestion. The resolution of Figure 5 has been improved to allow detailed infos more readable.

Round 2

Reviewer 2 Report

The text in Figure 5 is still small.

Author Response

Thank you again for your comment.

In the submitted revised version (Microsoft Word Format), the authors have improved the resolution of Figure 5 (A) heatmap to allow a sharper font image when zooming-in in the word format. However, when checking the PDF format on the submission website, the text font became blurred. Here I upload the Figure 5 file (.tif) directly. It will be appreciated if the publisher can handle it for us.